# Social predictors of the transition from anomie to deviance in adolescence

**Emanuel Adrian Sârbu**[1], **Bogdan Nadolu**[2]*, **Remus Runcan**[3], **Mihaela Tomiţă**[2], **Florin Lazăr**[4]

**1** Faculty of Baptist Theology and Faculty of Sociology and Social Work, University of Bucharest, Bucharest, Romania, **2** Faculty of Sociology and Psychology, West University of Timişoara, Timisoara, Romania, **3** Faculty of Educational Science, Psychology and Social Work, Aurel Vlaicu, University of Arad, Arad, Romania, **4** Faculty of Sociology and Social Work, University of Bucharest, Bucharest, Romania

* bogdan.nadolu@e-uvt.ro

## Abstract

Adolescence is a complicated, full of challenges and explorations period in life on the way to adulthood. The behaviour of adolescents is considerably re-configuring under the pressure of biological, psychological, and social transformations, and the internalization of community rules and values, as well as the adoption of desirable behaviours, is not always easy or successful. During adolescence, anomie can easily become an attractive *status quo*, but it can also evolve, however, relatively easy, to delinquency. This exploratory study, part of the Planet Youth project, is based on an analysis of 17 items from a questionnaire applied to a sample of 2,694 young people in Bucharest, Romania, in 2018, high schoolers in grades 9–11. The main objective of this approach was to assess the impact of some socio-cultural factors regarding school, family, peer group, and neighbourhood on the adoption of deviant and delinquent behaviours among Bucharest teenagers. For data analysis, two dependent variables were built by aggregating items in the questionnaire: the level of anomie (composed of 8 items) and deviant behaviour (composed of 17 items). As independent variables, 17 predictors composed from 67 questions from the questionnaire were used. The main results reflect a high level of anomie among the adolescents of Bucharest and a low level of deviance, and a weak link between these two variables. On the other hand, adolescent anomie and deviance are favoured by anger management, perceived peer attitudes to substance use and digital leisure, together with low parental surveillance.

## Introduction

In a society with an increasingly intense lifestyle, connecting to symbolic norms is becoming more and more of a challenge, especially since value systems are also coming under constant pressure from a range of alternative models. Adolescents, perhaps more than any other age range, are exposed both to extremely demanding existential incongruities, with a plethora of "opportunities" being offered by rival socialising agents, and to a multitude of "risks" masked by the promises of social success that these various models hold out–and all this at a time in life when they feel impelled to seek and find their optimal path in life.

**Data Availability Statement:** Due to internal and ethical rules and regulations, the whole data set is not to be made available for public sharing/ open access; however, the reasonable requests for data

will be analysed and other interested researchers may receive data like: the values behind the means, standard deviations and other measures reported; the values used to build graphs, labels etc. Data access should be required and motivated to the financing body: Icelandic Center for Social Research and Analysis - attn. to mr. Jon Sigfusson, Chairman of the Board, e-mail: jon@rannsoknir.is, or to dr. Emanuel Adrian Sarbu, Regional Representative Central-Europe, easarbu@ftb. unibuc.ro.

**Funding:** The publishing of this article was supported by West University of Timisoara, Romania. The funders had no role in study design, data collection and analysis, decision to publish, or preparation of the manuscript.

**Competing interests:** The authors have declared that no competing interests exists.

According to Smith & Bohm [1], anomie theories are aspects of alienation theory, since the dominant dimension of the former is normlessness = "the lack of any relevant norms or standards" [2], one of the five dimensions of the latter [3].

In psychiatry, alienation is defined as "a state of depersonalization or loss of identity in which the self seems unreal, thought to be caused by difficulties in relating to society and the resulting prolonged inhibition of emotion" [2]. An alienated person (i.e., a person "experiencing or inducing feelings of isolation or estrangement" [2]) may show one or more of the following symptoms [4]: fatigue; feeling different from everyone else; feeling distanced from work, family, and friends; feeling helpless; feeling left out of conversations or events; feeling separate from everyone else; feeling that the world is empty or meaningless; feeling unsafe when interacting with others; having a poor appetite; having difficulty approaching and speaking with others, especially with parents; feeling hopeless; insomnia; low self-esteem; overeating; refusing to obey rules; or sleeping excessively.

Smetana focused on the transition of American adolescents from childhood to adolescences and found that it included "minor but persistent conflict with parents over the everyday details of family life" and that family conflict rarely occurs over subjects such as drugs, politics, religion, and sex (maybe because of family members' reluctance to discuss such sensitive issues), but rather over issues of rule breaking and non-compliance with parental requests in such areas as choice of activities, friends and social life, disobedience, failure to finish tasks, fighting with siblings, home chores, personal hygiene, schoolwork, and teasing siblings. She also found that family conflicts "typically occurred over parental expectations rather than explicit rules, although families with preadolescents were more likely to have conflicts over rule-governed issues than were all other families" [5].

Anomie ("lack of the usual social or ethical standards in an individual or group" [2]) is used, in sociology, as a general theoretical concept to denote social and normative dissolution, to describe a strictly macro-sociological condition of the market economy and its ideology dominating other social institutions [6,7], and as a social psychological condition [8].

Anomie manifests itself at both individual (as lack of exteriority and constraint) and social (as aggregation of the lack of exteriority and constraint) levels, with direct effects on psychological well-being and problem behaviours among adolescents.

According to Merton (1968), anomie is a form of behaviour manifested by people suffering from social strain, i.e., from a mismatch between culturally prescriptive means and socially prescriptive goals, which makes them channel strain in different ways and, consequently, manifest different forms of anomic behaviour. The discrepancy between cultural premises and structural realities undermines social support for and promotes violations of institutional norms. An individual's possible adaptations to the discrepancy between culture and social structure (perceived as environmental pressure) are shown in Table 1 [9]:

If not dealt with properly, anomie in adolescents is a predictor of deviant behaviour ("activity that is proscribed by custom, social mores, or laws intended to curb or discourage such activity", "any behaviour considered to be grossly abnormal" [10]; academic dishonesty [11]; aggression / violence [12,13]; bullying ("seeking to harm, intimidate, or coerce someone perceived as vulnerable") [2,8]; conduct disorders (conduct disorder, intermittent explosive disorder, kleptomania, oppositional defiant disorder, pyromania) [14–17]; criminal activity [1,18–27]; depression [8]; Internet addiction [28,29]; poor school performance [30,31]; sexually transmitted diseases [12]; substance use: alcohol [12,22,32–35], drugs [8,12,22,36–38], marijuana [33], medicines, tobacco [8,33,37,39], water pipe tobacco [40]; suicide ideation [8,12,22,41,42]; teen pregnancy [43]; truancy ("staying away from school without good reason; absenteeism") [2,8,22,44,45]; excessive TV watching [8]; and vandalism [12].

**Table 1. Type of adaptations to the discrepancy between culture and social structure.**

| Type of adaptation | Cultural goal | Institutionalised means |
|---|---|---|
| **Conformity**: the individual continues to engage in legitimate occupational or educational roles despite environmental pressures toward deviant behaviour | acceptance | acceptance |
| **Innovation**: the individual has assimilated the cultural emphasis on the goal without equally internalizing the institutional norms | acceptance | rejection |
| **Ritualism**: the individual is an over conformist | rejection | acceptance |
| **Retreatism**: the individual has completely escaped from the pressures and demands of organized society | rejection | rejection |
| **Rebellion**: the individual publicly acknowledges his/her intention to change the norms and the social structure that they support in the interests of building a better, more just society | *rejection of prevailing goal or means and substitution of new goal or means* | *rejection of prevailing goal or means and substitution of new goal or means* |

After Merton, 1968, 195–211.

Adolescent anomie has been studied by Garfield, who reviewed over 50 years of work on anomie [36]; Vega *et al.* (1993), who studied risk factors for early adolescent drug use in ethnic and racial groups [46]; Bjarnason (1994), who found that suicidal behaviour is most strongly affected by psychological support from family and by suicide suggestion, with depression as intervening variable [41]; Bjarnason (1998), who found that parental support and religious participation increase perceived exteriority and constraint of the social world [32]; La Greca & Lopez (1998), who found that social anxiety is related with peer relations and friendships [47]; Mullan Harris, Furstenberg & Marmer (1998), who found beneficial effects for children of father's involvement on educational and economic attainment, delinquent behaviour, and psychological well-being [48]; Desforges & Abouchaar (2003), who approached the issue of parental involvement and support on pupil achievements and adjustment [49]; Thorlindsson & Bernburg (2004), who found that both community and individual level of social integration indicators have negative effects on adolescent delinquency, that the experience of anomie mediates a substantial part of these effects, and that the multi-level context of social integration and anomie moderates the effect of imitation (peer delinquency) on delinquent behaviour [50]; Smith & Bohm (2008), who found that crime/delinquency is a function of alienation [1]; Bjarnason (2009), who found that exteriority (i.e., experiencing the social world as an objective, predictable reality) is associated with more depression and less self-esteem, that constraint (i.e., the extent to which one experiences a personal commitment to the demands and expectations of society) is associated with a lower probability of daily smoking, illicit drug use, truancy, and suicide attempt, and that societal anomie is also associated with higher baseline levels of depression, self-esteem issues, and illicit drug use [8]; Selfhout *et al.* (2009), who established a relationship between different types of Internet use, depression, and social anxiety [51]; Polgar-Matthews (2011), who found that anomic condition during adolescence is positively related to adolescents' levels of aggression [13]; Kok & Goh (2012), who found that Malaysian youth indicated life was self-determined, and that this revealed the changing values among teenagers, which might be contributing to the high suicide rate in that country [42]; McCormick *et al.* (2013), who investigated the relationship between parent involvement, emotional support, and behaviour problems from an ecological perspective [52]; İçellioğlua & Özden (2014), who found that cyberbullying (as a new kind of peer bullying) is related with aggression and social anxiety in adolescents and young people [53]; Ļevina, Mārtinsone & Kamerāde (2015), who found that there was a significant difference in multidimensional anomie between Latvians belonging to different age groups (including adolescents) [54]; Runcan (2015), who approached Facebookmania as a psychic addiction to Facebook and its incidence

on the Z Generation [55]; Bashir & Singh (2018), who found that there is a significant positive relationship between anomie and academic dishonesty in college students [11]; and Cabrera *et al.* (2000), who analysed fatherhood in the 21st century [56].

In Romania, after the political change from December 1989, juvenile delinquency has significantly increased due to structural transformation of the entire social system. Increased delinquent behaviour of the teenagers has recorded not only an increase in frequency but also a decrease of the offenders' age. After the 2000s, the parents' working abroad has generated a new risk group for the children in this situation. The state recognition of this issue is still delaying and, thus, public policies and interventions for the protection of this category of teenagers are almost completely absent. Family and schools remain the main agents of socialization to ensure a good social integration of the future adults, but their task is more difficult in the context of easier access to various alternative models with a different values scale [57,58].

## 2. Materials and methods

In 2018, as part of the Planet Youth project, a large-scale sociological survey of the lives of young people in Romania was carried out. This involved an extensive omnibus questionnaire (with 296 questions) and a representative sample of 2,953 high schoolers from grades 9–11 from 78 high schools and colleges in Bucharest. Valid questionnaires were obtained from 2,694 subjects (48.4% boys and 51.6% girls) of 2,953 students in the sample, after data cleaning.

In accordance with the research protocol, initiated and coordinated by the Icelandic Centre for Social Research and Analysis (ICSRA), Reykjavik University, the local coordinator in Bucharest (the General Directorate of Social Welfare, Bucharest Municipality) signed a partnership agreement with all participating schools. To comply with the Romanian national rules and regulations and with the ethical requirements of the National Bioethics Committee of Iceland, great attention was paid in all stages of the program to guaranteeing the anonymity of participants and the confidentiality of their answers [59].

Before starting the data collection process, more than 3500 informative letters were sent to ALL the parents / tutors of the students in the sample, presenting the research and asking for their consent. Passive consent was obtained for the selected sample; the students who returned the letters signed by their parents who rejected their participation in the survey were not included, as instructed. The whole process was approved by the General Directorate of Social Welfare—Bucharest Municipality and by The School Inspectorate—Bucharest

The questionnaire is based on the Youth in Europe (Planet Youth) Program initiated in Iceland. In all participant cities, the same core-questionnaire and the same methodology was applied to ensure data comparison and replicability. In Bucharest, the program was implemented in 2008, under the coordination of The General Directorate of Social Welfare, and after signing a protocol with both The School Inspectorate of Bucharest Municipality and each participant school [59].

The Planet Youth questionnaire has been built with the intention to facilitate the understanding of the social circumstances of adolescent lives which can be connected with substance use, aiming to identify the risk and protective factors. For this reason, numerous indicators and variables have been included–measuring not only substance use, but also other factors like general wellbeing, relationship with peers and family, health, depressive mood, anger/ aggressiveness, suicidal behaviour, etc. The questionnaire is not intended to be similar to a clinical tool like SCL-90 or SCL-90 R; however, it is based on–and contains items–from validated scales like SCL-90, Beck's, Hamilton's, and others considered relevant.

The questionnaire was self-completed (pen/pencil and paper) by the students in the selected classes. After completing the questionnaire, each student placed it in the envelope provided, sealed it, and handed it to the research team representative in their school.

## Measures

The main research questions of the paper aimed at identifying the impact of several social factors related to school, family, peer group, and neighbourhood on the adoption of deviant, delinquent behaviour by Bucharest teenagers. To do so, a multiple linear regression model was applied, with deviant behaviour (DEV) as dependent variable grouping the results from 17 items. As independent variables, 16 variables generated on the basis of 66 items related to several social factors that can be associated with deviant and delinquent behaviour were used: school activity, relationship with parents, peer group, and neighbourhood, and social media use. From the factual data, only sex (as dummy variable) was used, because age was too close (most subjects were 15–16 years old). These predictors are explained below and the dependent variable is detailed in the Results section.

**Absenteeism risk** (summative q16a-c)–the cumulative score for the number of days that the subject did not attend school during the previous month for a range of reasons. Three items were included–for health reasons, truancy, and for other reasons–with answers from none to 5–6 days on a 5-step scale (minimum = 3 for no absences, maximum = 15 for 15 to 18 days of absence, median = 5).

**Commitment to study** (summative q17 a, b, c, d, j)–the cumulative score of five items about perceptions of the usefulness of school and student's own school status and performance [46]. The items involved use an inverse five step scale, from 1 = all the time to 5 = never, so the lowest score represents a low level of commitment to study (min = 1 due to non-answers maxim = 25, median = 18).

**Emotional wellbeing in school** (summative Q17 g, h, i)–represents a cumulative score for emotional approach to educational activities, based on three items with a five-point scale, from 1 = all the time to 5 = never), so that the highest score represents a high level of emotional wellbeing in school (minimum = 1 due to non-answers, maximum = 15, median = 13) [50].

**Parental support** (Q19)–the cumulative score for four types of parental support (care and affection, discussions about personal issues, study tips, support for various problems), each of these on a four-point scale, from 1 = very difficult to 4 = very easy), so that the highest score represent the highest level of parental support (min = 5, max = 20, median = 18) [60].

**Time spent with parents** (Q21)–the cumulative score from two items related to time spent with parents during weekdays and during the weekend, with a 5-point scale from 1 = almost never to 5 = almost always (min = 1, max = 10, median = 7) [61].

**Parental rule setting** (Q24 b, c, d)–the cumulative score of three items, each of them on a four-point scale, from 1 = it applies very well to me to 4 = it applies very little to me, so that the highest score represents the lowest level of parental monitoring (min = 1, max = 12, median = 8) [62].

**Parental monitoring** (sum Q24 e, f, k)–the cumulative score of three items, each of them on a four-point scale, from 1 = it applies very well to me to 4 = it applies very little to me, so that the highest score represents the lowest level of parental monitoring (min = 1, max = 12, median = 4) [62].

**Relative deprivation of family** (Q27)–the cumulative score of four items, each of them on a five points scale, from 1 = almost never to 5 = almost always, so that the highest score represents the highest level of family deprivation (min = 1, max = 20, median = 5).

**Intergenerational closure/social capital** (sum Q24 g, h, i, j)–the connections between one's own parents and the parents of friends, on a similar scale to that above (min = 1, max = 16, median = 11) [62].

**Personal safety** at home, at school, in the neighbourhood I live in (Q23)–a sum of three items with a five-point scale from 1 = almost never to 5 = almost always (min = 1, max = 15, median = 13).

**Neighbourhood social networks** (Q25)–the cumulative score of six items, each on a five-point scale, from 1 = always to 5 = never), so that the highest score represents the weakest level of neighbourhood social networks (min = 1, max = 30, median = 20).

**Peer support** (Q20)–the cumulative score of the five types of peer support, just as the previous question, each on a four-point scale, from 1 = very difficult to 4 = very easy), so that the highest score represents the highest level of peer support (min = 1, max = 20, median = 16) [63].

**Perceived peer attitudes to substance abuse** (Q28)–the cumulative score of four-items, each on a four-point scale, from 1 = totally agree to 4 = totally disagree, so that the highest score represents the highest level of perceived peer attitudes against substance abuse (min = 1, max = 16, median = 16).

**Anger management/control problems** (Q30)–the cumulative score of five items, each on a four-point scale, from 1 = almost never to 4 = always, so that the highest score represents the lowest level of anger management (min = 1, max = 20, median = 10) [64].

**Level of anomie** (Q35) as acceptance of the rules–the cumulative score of eight items, each on a four-point scale, from 1 = totally agree, to 4 = totally agree that the highest score represent the lowest level of anomie. To include all the cases with at least one answer, this score was divided by the number of items different from zero.

**Digital leisure** (Q75 a-c)–the cumulative score of three items related to movies, TV, PC games and social-media, each on eight-point scale, from 1 = almost never to 8 = 6 hours daily or more, so that the highest score represent highest digital leisure daily consume.

## 3. Results

The socio-demographic profile of the sample is presented in Table 2.

Table 3 presents descriptive statistics of the scales used as independent variables (their meanings min/max are presented under Methods).

Concerning the level of anomie for Bucharest teenagers, this scale was formed by eight items and the frequency of the answers are presented in Table 4.

As can be observed in Fig 1, there are quite high manifestations of the level of anomie for Bucharest teenagers, for all included items the general score for agreeing to various form of breaking the rules getting higher value than the disagree with these.

For a deeper analysis of the anomic level, a new variable as an anomie index was calculated by dividing the sum of the answers to all these eight items to the number of items that were different from zero. Thus, the new variable gets values from 1 to 5 that can be grouped on a 4-step scale as it is presented in Fig 2.

Thus, the existence of a very high level of anomie among the teenagers from Bucharest was noted, over 66% of them showing at least the willingness to violate the norms. This distribution characterizes both boys and girls equally (test t = 1.858, sig<0.063), no matter their age (Anova F test = 1.970, sig < 0.080), grade (Anova F test = 2.531, sig < 0.080) or family type (Anova F test = 1.558, sig < 0.122). Instead, there are significant differences between parents' training levels, adolescents' orientation towards anomie being directly associable with a lower level of parents' education (for mothers, Anova test F = 2.362, sig < 0.028, for fathers, Anova test F = 3.972, sig < 0.001). The correlations between the parents' level of training and the tendency towards anomie, although reflecting a weak link, are also statistically significant (for mothers, r = -0.068, sig < 0.001, for fathers, r = -0.094, sig < 0.001). Other statistically

**Table 2. Socio-demographic profile of the sample.**

| Sex | boy | girl | | | | |
|---|---|---|---|---|---|---|
| | 48.4% | 51.6% | | | | |
| Year of birth | 2000 | 2001 | 2002 | 2003 | 2004 | 2005 |
| | 1% | 7.3% | 72.2% | 18.8% | 0.6 | 0.1% |
| Grade | 9th grade | 10th grade | 11th grade | | | |
| | 2.5% | 93.6% | 3.9% | | | |
| Family structure | I live with my mother and father equally | | | 69% | | |
| | Only one parent | | | 14.5% | | |
| | One parent and his/her partner | | | 5.3% | | |
| | Extended family (with grandparents and with/without one parent) | | | 9.2% | | |
| | Other | | | 2% | | |
| Highest level of schooling | | | | mother | father | |
| | Graduated from university | | | 50,4 | 44,5 | |
| | Started university but has not finished | | | 3,3 | 3,8 | |
| | Graduated from junior college or trade school | | | 8,3 | 9,4 | |
| | Started junior college but has not finished | | | 1,5 | 1,8 | |
| | Graduated from high school | | | 25,1 | 29,8 | |
| | Started high school but has not finished | | | 8,3 | 7,8 | |
| | Primary school or less | | | 3,0 | 2,9 | |

Source: Authors work.

significant correlations have been recorded between anomie and commitment to studies (r = 0.227, sig < 0.001), anger management (r = -0.171, sig < 0.001), emotional wellbeing in school (r = 0.152, sig < 0.001), perceived peer attitudes to substance use (r = 0.137, sig < 0.001), absenteeism risk (r = -0.131, sig < 0.001), parental support (r = 0.127,

**Table 3. Descriptive statistics for applied scales.**

| Measure | N | Min | Max | Mean | SD | Cronbach Alpha |
|---|---|---|---|---|---|---|
| Absenteeism risk | 2111 | 3 | 15 | 5.47 | 2.585 | 0.500 |
| Commitment to studies | 2553 | 1 | 25 | 17.37 | 4.164 | 0.736 |
| Emotional wellbeing in school | 2616 | 1 | 15 | 12.36 | 2.686 | 0.647 |
| Parental support/control | 2623 | 1 | 20 | 16.90 | 3.315 | 0.844 |
| Time spent with parents | 2630 | 1 | 10 | 6.84 | 2.206 | 0.709 |
| Parental rule setting | 2612 | 2 | 12 | 8.05 | 2.448 | 0.743 |
| Parental monitoring | 2653 | 1 | 12 | 4.96 | 2.079 | 0.710 |
| Intergenerational closure/social capital | 2637 | 1 | 16 | 10.64 | 3.043 | 0.809 |
| Relative deprivation level of family | 2627 | 1 | 20 | 6.31 | 3.170 | 0.792 |
| Personal safety (at home, at school in the neighbourhood) | 2651 | 1 | 15 | 12.58 | 2.355 | 0.617 |
| Neighbourhood social networks | 2623 | 1 | 30 | 19.07 | 5.809 | 0.850 |
| Peer support | 2608 | 1 | 20 | 15.57 | 3.560 | 0.838 |
| Perceived peer attitudes to substance use | 2661 | 1 | 16 | 14.60 | 2.426 | 0.826 |
| Anger management/control problems | 2614 | 1 | 20 | 10.48 | 3.831 | 0.814 |
| Level of anomie | 2588 | 1 | 40 | 13,93 | 6,314 | 0.746 |
| Digital leisure | 2566 | 3 | 24 | 12.30 | 4.587 | 0.447 |

Source: Authors work.

**Table 4. Items related to the level of anomie.**

| Q35 Agree or disagree: | Strongly agree | Agree somewhat | I don´t know | Disagree somewhat | Strongly disagree | DK/NA |
|---|---|---|---|---|---|---|
| One can break most rules if they don´t seem to apply | 12.9 | 28.4 | 28.9 | 17.9 | 10.4 | 1.6 |
| I follow whatever rules I want to follow | 32.4 | 30.6 | 11.5 | 15.3 | 8.5 | 1.7 |
| In fact, there are very few absolute rules in life | 20.5 | 28.6 | 32.4 | 10.6 | 6.3 | 1.6 |
| It is difficult to trust anything because everything changes | 36.6 | 35.4 | 14.7 | 8.8 | 2.7 | 1.7 |
| In fact, nobody knows what is expected of him / her in life | 30.4 | 31.7 | 20.5 | 10.7 | 4.9 | 1.7 |
| One can never be certain of anything in life | 35.9 | 33.1 | 11.9 | 11.4 | 6 | 1.6 |
| Sometimes one needs to break rules in order to succeed | 30.4 | 37.2 | 15 | 10.5 | 5.2 | 1.6 |
| Following rules does not ensure success | 29 | 26.1 | 19.8 | 14.8 | 8.5 | 1.8 |

sig < 0.001), personal safety at home, at school, in the neighbourhood (r = 0.125, sig < 0.001), digital leisure (r = 0.227, sig < 0.001), time spent with parents (r = 0.111, sig < 0.001), and parental monitoring (r = -0.097, sig < 0.001).

The next stage was the computation of a new variable, deviant behaviour, which can have values from 1 = non to 4 = deviant and delinquent, on the basis of the following questions: Q64. Offences committed in the last 12 months, Q66 Exercise of physical or sexual violence in the last 12 months, Q67 Group delinquency and Q69 Violent behaviour. Firstly, due to the significant size of the sample, it was considered relevant to present the descriptive statistics for each of these items in Tables 5–8:

Fig 3 contains a graphic representation of deviant and delinquent behaviours, involving the grouping together of all respondents who claimed at least one action from the category concerned.

On the basis of these 17 items, a new variable, deviant behaviour, was constructed by aggregating (summing up) the scores obtained for each question (from never = 1 point to 18 or more times = 7 points) are between 17 and 113 points (96 points length from minimal to maximal value). The total score obtained allows the grouping of the subjects into four distinct categories, presented in Table 9:

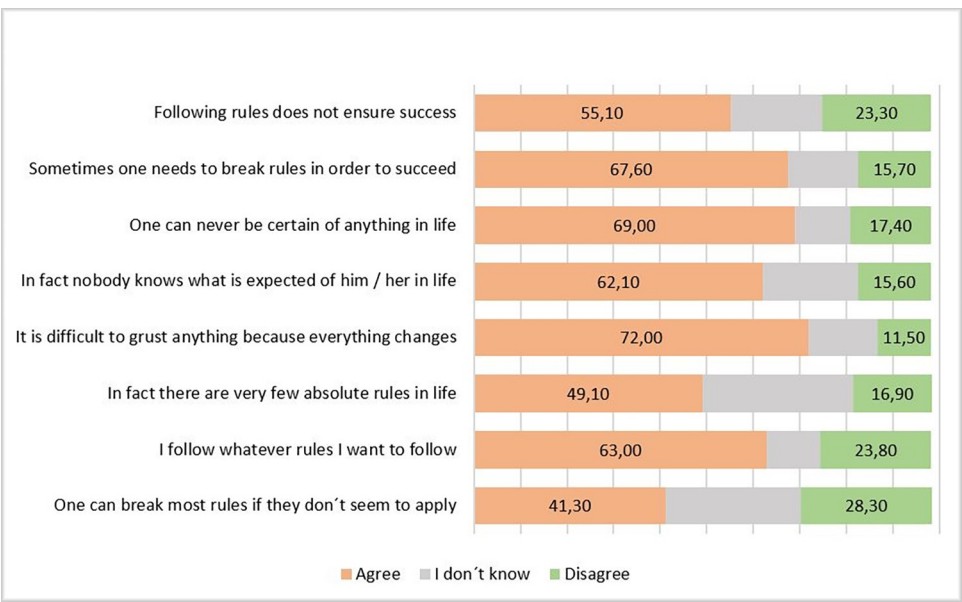

**Fig 1. Sentences about breaking/observing the rules.**

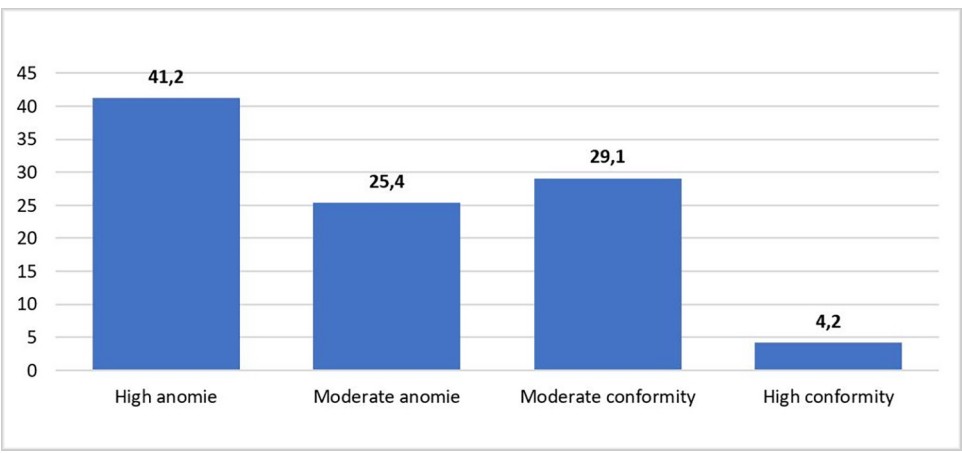

**Fig 2. Level of anomie.**

1 = non-deviance: did not commit any of the above activities (score up to 17 points);

2 = low deviance: cumulative score of deviant behaviours between 18 and 49 points;

3 = moderate deviance: cumulative score between 50 and 81 points;

4 = high deviance: cumulative score over 82 points.

**Table 5. How often during the last 12 months have you–delinquent behaviour.**

| Deviant behaviours... | Never | Once | 2–5 times | 6–9 times | 10–13 times | 14–17 times | 18 times or more | DK/NA |
|---|---|---|---|---|---|---|---|---|
| used physical violence in order to rob/steal | 94.6 | 0.8 | 0.4 | 0.4 | 0.1 | 0.1 | 0.7 | 2.9 |
| broken into a building or a car to steal | 94.5 | 0.7 | 0.4 | 0.2 | 0.3 | 0.1 | 0.7 | 3.1 |
| stolen something worth more than 3 normal movie tickets | 91.8 | 2.8 | 1 | 0.3 | 0.4 | | 0.9 | 2.8 |
| other offence | 89.5 | 2.9 | 2.2 | 0.7 | 0.1 | 0.2 | 1.3 | 3.1 |
| stolen something worth less than 3 normal movie tickets | 87.3 | 4.8 | 2.8 | 0.9 | 0.1 | 0.1 | 1 | 3 |
| damaged or vandalized things that did not belong to you | 84.3 | 6.4 | 3.7 | 0.7 | 0.4 | 0.1 | 1 | 3.4 |

Source: Authors work.

**Table 6. Violent behaviours–delinquent behaviour.**

| | Never | Once | 2–5 times | 6–9 times | 10–13 times | 14–17 times | 18 times or more | DK/NA |
|---|---|---|---|---|---|---|---|---|
| Have you exerted sexual violence during the last 12 months? | 94.7 | 0.9 | 0.4 | 0.1 | 0.2 | 0.1 | 0.5 | 3.1 |
| Have you exerted physical violence during the last 12 months? | 79.3 | 9.6 | 5.5 | 1.1 | 0.3 | 0.2 | 1.1 | 2.9 |

Source: Authors work.

**Table 7. Violent group behaviours (Q67).**

| Q67 How often during the last 12 months have you: | Never | Once | Twice | 3–4 times | 5 times or more often | DK/NA |
|---|---|---|---|---|---|---|
| Been a part of a group physically hurting an individual | 80.7 | 8.7 | 3.1 | 1.8 | 2.4 | 3.3 |
| Been a part of a group starting a fight with another group | 76.7 | 11.6 | 3.2 | 1.9 | 3.3 | 3.3 |
| Been a part of a group teasing an individual | 47 | 23.9 | 10.2 | 5 | 10.8 | 3 |

Source: Authors work.

**Table 8. Aggressive behaviours (Q69).**

| Q69 How often during the last 12 months: | Never | Once | 2–5 times | 6–9 times | 10–13 times | 14–17 times | 18 times or more | DK/NA |
|---|---|---|---|---|---|---|---|---|
| Held somebody by their neck | 79.9 | 6.6 | 3.9 | 1.7 | 0.8 | 0.5 | 2.7 | 4 |
| Knocked somebody over | 66.7 | 12.7 | 8.1 | 3.1 | 1.2 | 0.9 | 3.7 | 3.6 |
| Threatened somebody with violence | 63.3 | 11.8 | 8 | 3.4 | 1.9 | 1.1 | 6.5 | 4 |
| Kicked somebody | 59.3 | 15 | 10.1 | 3.7 | 2 | 1 | 4.9 | 4 |
| Punched somebody | 57.1 | 15.1 | 10.8 | 3.7 | 1.7 | 0.8 | 7.2 | 3.6 |
| Hit/slapped somebody | 48.2 | 18.2 | 13.2 | 5.9 | 2.5 | 1.6 | 6.6 | 3.8 |

Source: Authors work.

The analysis of these distributions from the perspective of the scales used in the questionnaire is presented in Table 10.

On the basis of these values, it is quite obvious that two very clear groups emerge: the non-deviant subjects (the first category), and the high-deviant subjects (the fourth category). Thus, the non-deviant subjects have a low level of absenteeism risk (mean = 4.79 from 15), a higher commitment to their studies (mean = 18.8 from 25), higher levels of emotional wellbeing in school (mean = 12.74 from 15), receive stronger parental support (mean = 17.42 from 20), spend more time with parents (mean = 7.33 from 10), and have the great sense of personal safety at home and school (mean = 12.96 from 15). For these teenagers, parents have a stronger

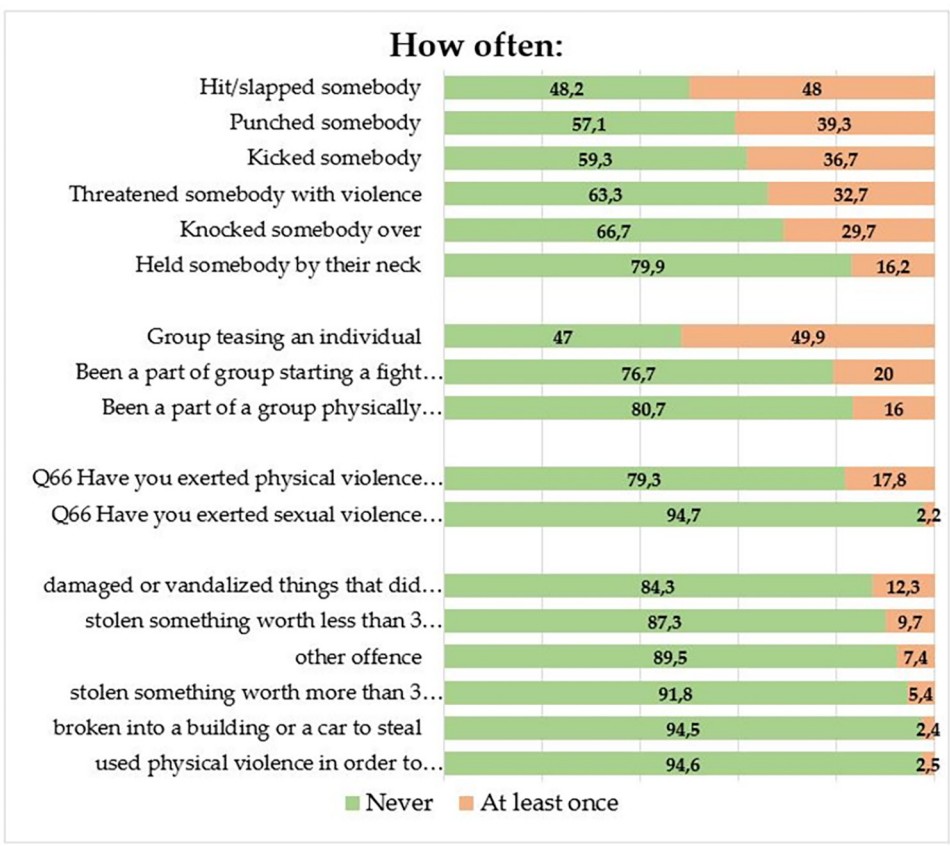

**Fig 3. Deviant vs. Non-deviant behaviours.**

**Table 9. Types of deviant behaviour (new aggregate variable).**

|  | Frequency | Percentage | Maximum | Mean | Std. Deviation |
|---|---|---|---|---|---|
| 1 non-deviance | 632 | 23.5 | 17 | 16.63 | 1.685 |
| 2 low deviance | 1845 | 68.5 | 49 | 23 | 7.274 |
| 3 moderate deviance | 120 | 4.5 | 81 | 59.33 | 7.360 |
| 4 high deviance | 17 | 0.6 | 113 | 98.59 | 9.792 |
| System Missing | 80 | 3.0 |  |  |  |
| TOTAL | 2694 | 100.0 |  |  |  |

Source: Authors work.

rule setting (mean = 7.92 from 12) and monitoring role (mean = 4.45 from 12), and they have a smaller intergenerational distance (mean = 10.4 from 16), a better financial situation (mean = 5.92 from 20, that means the highest level of family deprivation), better anger management (mean = 8.81 from 20) and a greater degree of independence of peer attitudes to substance use (mean 15.60 from 16, that means peer attitudes against substance use). In contrast with this, the high-deviant subjects have a higher level of absenteeism risk (9.07), the lowest level of parental control (8.17), wider intergenerational distance (11.23), the worst family financial situation (7.37), the worst anger management (14.00 from 20) and the highest tolerance of the peer attitudes to substance use (8.11). They also spend more time daily on social-media (15.93 from 24). Nonparametric test Kruskal Wallis of these associations shows significant statistical differences for almost all items, except for peer support and intergenerational closure as shown in Table 11.

According to these data, all analysed scales, except peer support and intergenerational closure/social capital have recorded a significant statistical difference among the four levels of deviance and they can act as potential predictors for the delinquent behaviour of the Bucharest teenagers. To evaluate these causalities, a multiple linear regression model, with the 16 scales

**Table 10. Type of deviant behaviour vs. scales results.**

| Type of deviant behaviour (mean) | 1 non-deviance | 2 low deviance | 3 moderate deviance | 4 high deviance | TOTAL sample |
|---|---|---|---|---|---|
| Absenteeism risk | 4.79 | 5.52 | 7.16 | **9.07** | 5.44 |
| Commitment to studies | **18.8** | 17.22 | 14.46 | 10.88 | 17.43 |
| Emotional wellbeing in school | **12.74** | 12.41 | 10.73 | 8.64 | 12.39 |
| Parental support | **17.42** | 16.83 | 16.19 | 14.00 | 16.92 |
| Peer support | **15.53** | 15.63 | **15.16** | 14.76 | 15.58 |
| Time spent with parents | **7.33** | 6.74 | 6.12 | 5.11 | 6.84 |
| Personal safety at home, at school, and in the neighbourhood, I live in | **12.96** | 12.55 | 12.10 | 10.29 | 12.61 |
| Parental rule setting | 7.92 | 8.06 | **8.56** | 8.00 | 8.05 |
| Parental monitoring | 4.45 | 5.02 | 6.04 | **8.17** | 4.95 |
| Intergenerational closure/social capital | 10.40 | 10.72 | 10.70 | **11.23** | 10.65 |
| Neighbourhood social networks | **19.56** | 19.05 | 17.80 | 16.47 | 19.10 |
| Relative deprivation—family | 5.92 | 6.41 | 6.43 | **7.37** | 6.30 |
| Perceived peer attitudes to substance use | **15.60** | 14.60 | 12.79 | 8.11 | 14.61 |
| Anger management/ control problems | 8.81 | 10.85 | **12.92** | 14.00 | 10.47 |
| Level of Anomie (min = high) | 15.25 | 13.82 | 12.25 | **8.76** | 14.06 |
| Digital leisure | 10.72 | 12.58 | **15.40** | 15.93 | 12.29 |

Source: Authors work.

**Table 11. Kruskal Wallis analysis of the deviant behaviours by scales (between groups).**

| Type of deviant behaviours by 16 Scales | $\chi^2$ | df | Sig. |
|---|---|---|---|
| Absenteeism risk | 72.789 | 3 | <0.001 |
| Commitment to studies | 174.117 | 3 | <0.001 |
| Emotional wellbeing in school | 66.538 | 3 | <0.001 |
| Parental support | 40.531 | 3 | <0.001 |
| *Peer support* | *0.638* | *3* | *0.888* |
| Time spent with parents | 60.607 | 3 | <0.001 |
| Personal safety at home, at school, and in the neighbourhood, I live in | 33.912 | 3 | < .001 |
| Parental rule setting | 9.440 | 3 | 0.024 |
| Parental monitoring | 87.417 | 3 | < .001 |
| *Intergenerational closure/social capital* | *5.917* | *3* | *0.116* |
| Neighbourhood social networks | 11.597 | 3 | 0.009 |
| Relative deprivation—family | 21.487 | 3 | <0.001 |
| Perceived peer attitudes to substance use | 116.444 | 3 | <0.001 |
| Anger management /control problems | 203.789 | 3 | <0.001 |
| Level of anomie | 47.092 | 3 | <0.001 |
| Digital leisure | 135.227 | 3 | <0.001 |

Source: Authors work.

and sex as independent variables and the delinquency score as dependent variable was applied. From these, only twelve variables are representative for the predicting 31% of the deviant behaviour ($r^2 = 0.318$, F = 76.008, sig < 0.001), as shown in Table 12.

To note the significant difference in the sex variable, boys being more involved in deviant behaviours than girls. According to this model, the factors with the highest impact on deviant behaviour are anger management/control problems (β = 0.216), perceived peer attitudes to substance use (β = -0.198), sex (β = 0.183) and digital leisure (β = 0.130). At the opposite pole, the factors with the least impact are peer support (β = 0.049), time spent with parents (β = -0.042), and level of anomie (β = -0.040).

## 4. Discussion

Following the analysis of these data, the poor interconnection between anomie and deviance was remarked: there is a statistically significant correlation, but of low value (r = -0.112,

**Table 12. Multilinear regression analysis for Deviant behaviour.**

| Variable | *b* | β |
|---|---|---|
| Commitment to studies | -0.255 | -0.085** |
| Perceived peer attitudes to substance use | -0.993 | -0.198** |
| Anger management/control problems | 0.674 | 0.216** |
| Sex (0 = girl) | 4.286 | 0.183** |
| Digital leisure | 0.337 | 0.130** |
| Parental monitoring | 0.663 | 0.114** |
| Absenteeism risk | 0.395 | 0.087** |
| Neighbourhood social networks | -0.104 | -0.051** |
| Peer support | 0.169 | 0.049** |
| Emotional wellbeing in school | -0.233 | -0.051** |
| Time spent with parents | -0.226 | -0.042** |
| Level of anomie | -0.580 | -0.040** |

sig < 0.001). Strictly as a mean value, the anomie level of the Bucharest teenagers is 65% (total inverted and transformed average score) while deviance is 22% (the average value of the compound variable, transformed into percentage). The deviance level is based on the teenager's answers to some dedicated questions (a minimal alteration can be assumed under the specific tendency to boast).

Delinquent behaviour is predominantly marginal: sexual violence (2.2% at least once), used physical violence to steal or rob (2.5% at least once), breaking into a building or a car to steal (2.4%), stealing something worth more than three standard movie tickets (5.4%) or less than three standard movie tickets (9,7%), damaging or vandalizing things that do not belong to you (12,3%). At the same time, 23.5% of the teenagers from Bucharest have not previously committed any of the 17 acts of deviation assessed. Thus, although there is an accentuated temptation of adolescents in Bucharest to oppose the norms, the level of adoption of deviant behaviours is still very low. At national level, the number of minors definitively convicted by the courts decreased from 1983 persons in 1990 to 828 persons in 2018 (of which 567 with custodial measures, that is, for serious offence). The share of adolescents convicted in 2018 represents only 0.02% of the total minor population in Romania [65].

According to the analysed data, anomie is equally distributed among boys and girls, being favoured by the low level of parental education, by the interest in school, by absenteeism and wellbeing in school, by anger management, by the time spent with parents, by parental support and monitoring, by the safety of the environment in which they live, by the peers' tolerance of substance use and, last but not least, by the time spent online in leisure activities. Among these factors, according to the applied multiple linear regression model, anger management, perceived peer attitudes to substance use, sex, and digital leisure are factors with increased impact on deviant behaviour. These results are convergent with other similar studies applied on teenage population [66–69]. The intensive use of mobile phones without adult surveillance and, implicitly, the access to various social media platforms that promote plenty of alternative behaviours will remain a significant source for developing anomic attitudes and even deviant behaviours. Concerning the influence of the other macro-economic and social factors, in Romania, in 2018, there was a quite stable situation, without any significant disturbance (it was the year before the pandemic Covid-19 begun).

The non-deviant young people in our study have adopted a classic sustainable lifestyle: they are interested in school and they have good support from their families, a positive social context and a good level of anger management. At the opposite end of the spectrum, delinquent youth are not integrated in school, they have serious problems with their parents, including deprivation and low levels of surveillance, an insecure social context, and problems with anger management. Peer support has a comparable impact on all types of behaviour, so a good social environment at school has a direct positive effect, while a bad one has a negative impact. In addition, the amount of time spent online daily is also a valid predictor for anomic attitude and deviant behaviour, especially if there is a lack of supervision from the adults.

## Implications, limitations and future directions

The present findings suggest that, even if the level of anomie is high among Bucharest teenagers, their deviant behaviour remains at a low level (92% are actually non-deviance or low deviance experience). The presence of three factors with high impact on both aspects, anomie level and deviance level–anger management, peer attitude and digital leisure–represents a real risk, especially for boys, of crossing the line from attitude to behaviour, and it can have very serious consequences. The positive social context offered by the school and by the family represent, probably, the easier way to avoid these transformations.

There are several limitations of this interpretation that must be noted. First of all, the items related to deviant behaviour are very sensible *per se*, and for teenagers it is very likely to avoid a complete sincerely answer because of social desirability, or, on the contrary, to try to boast and exaggerate. The regression model has a quite limited level of explanation, only 31%, even though 17 distinct scales were used, which theoretically should have covered the entire topic of deviant behaviour. It is very possible to have other predictors with a higher impact that were not included in the analysis (acting under influence, or disturbing the macro social context, for example). Of course, a longitudinal analysis for two or three waves will be more appropriate for a better understanding of the transition process (but without including the pandemic period that has affected all teenagers' behaviour).

## Acknowledgments

Special thanks to Stuart and Dorothy Elford for proof reading in English, as well as for their constructive feedback regarding specific aspects of the research methodology.

## Author Contributions

**Conceptualization:** Emanuel Adrian Sârbu, Florin Lazăr.

**Data curation:** Bogdan Nadolu.

**Formal analysis:** Bogdan Nadolu, Remus Runcan.

**Funding acquisition:** Mihaela Tomiţă.

**Investigation:** Emanuel Adrian Sârbu.

**Methodology:** Emanuel Adrian Sârbu, Florin Lazăr.

**Project administration:** Remus Runcan.

**Resources:** Mihaela Tomiţă.

**Software:** Bogdan Nadolu, Florin Lazăr.

**Supervision:** Remus Runcan, Mihaela Tomiţă.

**Validation:** Emanuel Adrian Sârbu, Bogdan Nadolu, Florin Lazăr.

**Visualization:** Bogdan Nadolu.

**Writing – original draft:** Bogdan Nadolu, Remus Runcan.

**Writing – review & editing:** Emanuel Adrian Sârbu, Bogdan Nadolu, Remus Runcan, Mihaela Tomiţă.

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
