## [Decision Letter · Decision Letter 0]

2 Feb 2022

PONE-D-21-27300Transition from anomie to delinquency in adolescence: profiles and associated factorsPLOS ONE

Dear Dr. NADOLU,

Thank you for submitting your manuscript to PLOS ONE. After careful consideration, we feel that it has merit but does not fully meet PLOS ONE’s publication criteria as it currently stands. Therefore, we invite you to submit a revised version of the manuscript that addresses the points raised during the review process.

Although the first reviewer suggested the manuscript be rejected due to the difficulty in evaluating the methodology, both reviewers highlighted the strengths of the study and its value. So, instead of rejecting it, the authors have an opportunity to rework the manuscript and do the extensive rewriting the first reviewer indicates. In that case, it could undergo a second round of reviews.Please ensure that your decision is justified on PLOS ONE’s publication criteria and not, for example, on novelty or perceived impact.

We look forward to receiving your revised manuscript.

Kind regards,

Pedro Vieira da Silva Magalhaes, M.D., Ph.D.

Academic Editor

PLOS ONE

https://journals.plos.org/plosone/s/file?id=ba62/PLOSOne_formatting_sample_title_authors_affiliations.pdf"

2. Please improve statistical reporting and refer to p-values as "p<.001" instead of "p=.000". Our statistical reporting guidelines are available at https://journals.plos.org/plosone/s/submission-guidelines#loc-statistical-reporting.

“The publishing of this article was supported by West University of Timisoara, International Centre for Interdisciplinary Research on Human Rights, Timisoara, Romania.”

5. Thank you for stating the following in the Funding Section of your manuscript:

“The publishing of this article was supported by West University of Timisoara, International Centre for Interdisciplinary Research on Human Rights, Timisoara, Romania.”

“The publishing of this article was supported by West University of Timisoara, International Centre for Interdisciplinary Research on Human Rights, Timisoara, Romania.”

7. We note that you have indicated that data from this study are available upon request. PLOS only allows data to be available upon request if there are legal or ethical restrictions on sharing data publicly. For more information on unacceptable data access restrictions, please see http://journals.plos.org/plosone/s/data-availability#loc-unacceptable-data-access-restrictions.

8. Please include your full ethics statement in the ‘Methods’ section of your manuscript file. In your statement, please include the full name of the IRB or ethics committee who approved or waived your study, as well as whether or not you obtained informed written or verbal consent. If consent was waived for your study, please include this information in your statement as well.

Reviewers' comments:

Reviewer's Responses to Questions

**Comments to the Author**

1. Is the manuscript technically sound, and do the data support the conclusions?

Reviewer #1: No

Reviewer #2: Partly

2. Has the statistical analysis been performed appropriately and rigorously? 

Reviewer #1: No

Reviewer #2: Yes

3. Have the authors made all data underlying the findings in their manuscript fully available?

Reviewer #1: No

Reviewer #2: Yes

4. Is the manuscript presented in an intelligible fashion and written in standard English?

Reviewer #1: Yes

Reviewer #2: Yes

5. Review Comments to the Author

Reviewer #1: The research theme is relevant and in line with what is produced internationally in the area. However, the manuscript has serious problems, as indicated below, that make it impossible to fully evaluate (for example, poorly explained method) and its suggestion for publication. Some main points are presented below. However, due to the importance of the study and the robustness of the sample size, it is suggested that it be reworked for future submission.

It is not clear in the abstract the statistical analyzes that were carried out and what was presented as the objective of the study ("identify socio-cultural factors that may operate predictors of alienation and of deviant or even delinquent behavior"), is not aligned with what was presented as the main result ("Bucharest young people do not exhibit deviant behaviors or only to a minor degree, and that this is largely determined by a positive environment in terms of social relations, economic status, and level of culture.") . Furthermore, what is presented as the main result in the abstract is not congruent with the data presented in the manuscript, since 20% of the sample presented a type of behavior classified as “delinquency” and 12% as “high deviance” (Table 8).

In the introduction, the concepts of anomie and alienation are well defined and with a presentation of previous literature. Despite this, other variables that are addressed in the article (the risk factors assessed, such as parental monitoring, peer support, substance abuse, etc.) are not discussed based on the literature.

The method would require a characterization of the participants (age and gender) and a presentation of the psychometric parameters of the instrument adopted. In addition, more detail on the statistical analysis used would be necessary, describing the complete analysis process (data processing, choice of variables, descriptive statistics performed, procedure for carrying out logistic regression). As there is no description of the analysis procedure, it is not possible to know and to assess the rigor used in the process.

In the results, there is a presentation of definitions of scale variables and their composition that should be presented as “measures” in the method. There is no alignment between what is proposed to analyze the method and the analyzes presented in the results.

In the discussion, there is not enough discussion with literature data about the results found and the conclusions are not aligned with the results presented.

Thus, it is suggested to reject the scientific article since with little clarification of the method, a rigorous review is not possible. Furthermore, the conclusions and discussions are not aligned with the presented result.

Reviewer #2: Dear Authors,

First of all, I would like to emphasize the relevance of this study, which presents predictors of the transition from anomie to delinquency. Although the study was conducted at a local level, with young Europeans from Bucharest, Romania, it reflects the conditions experienced by many other populations and helps us to think micro- and macro-structurally about public policies that favor care and attention to situations of risk and delinquency.

I would like to list some points that drew my attention and that I believe deserve being reviewed aiming to improve the quality of this study:

1) I suggest that the title, “Transition from anomie to delinquency in adolescence: Profiles and associated factors”, be reconsidered, because the study does not exactly reflect the profile of young people, but rather the factors that associate the transition from anomie to delinquency.

2) In the Introduction, the authors refer to some studies that correlate the predictors of what could potentially trigger a transition from anomie to delinquency; however, none of them discuss the social or structural conditions that potentiate such situation. In developing countries, for instance, macroeconomic conditions prevail over individual and even community conditions regarding the experience of unruly and deviant adolescence. I believe it is important to emphasize this comparative aspect, which would strengthen the study and allow its scope to be expanded beyond subjective conditions and personal inabilities to comply or not with societal rules.

3) This study also sought to respond to the following question: “What are the most relevant factors associated with juvenile delinquency in Bucharest, Romania?” To this end, the authors resourced to a 17-item questionnaire. To better contextualize the research, I suggest that the authors include some information, if available, on the state of the art of juvenile delinquency in Romania and, in particular, in Bucharest, because only the data presented do not show whether delinquency is high or low among young Romanians.

4) The Methodology does not clearly present the profile of the study participants: there is need to explain, for example, the profile of the young people who responded to the questionnaire: age group, sex and gender identity, social class, ethnicity, in order to better understand the eligibility criteria of the study.

5) The Results and their statistical analysis faithfully demonstrate and answer the research question.

6) Regarding the Discussion, there is need for a more comprehensive analysis of both the results found and the findings of the literature that support the arguments. How does the theory of anomie, proposed by Merton (1968), assists us with thinking about the transition from anomie to delinquency in young Romanians? What pre-existing conditions do these young people have that motivate them to go beyond culturally and socially determined limits? Making good use of the studies and theories presented here could better support the evidence of the results.

7) In addition, the authors state that the amount of time spent on communication and information technologies (CIT) by today's youth emerges as a predictor of the high level of deviant behavior, but not necessarily of delinquent acts. To support this argument, it is important to review communication theories on the excessive and abusive use of CTI among young people.

Finally, I would like to say that I am grateful for having had the opportunity to read this study.

Sincerely,

6. PLOS authors have the option to publish the peer review history of their article (what does this mean?). If published, this will include your full peer review and any attached files.

Reviewer #1: No

Reviewer #2: No

---

## [Author Response · Author response to Decision Letter 0]

9 Apr 2022

Answers to Editor:

1. Please ensure that your manuscript meets PLOS ONE's style requirements, including those for file naming

We have made the following corrections/upgrades:

- included the line numbers

- double-spaced the paragraphs

- used the correct file name (Manuscript)

2. Please improve statistical reporting and refer to p-values as "p<.001" instead of "p=.000".

The p-values are reporting as “p<0.001”

We have had .000 only in two tables and we have changed it with <0.001, hope this is acceptable.

We have received funding only for the publishing fee, not for the research.

4. In your Data Availability statement, you have not specified where the minimal data set underlying the results described in your manuscript can be found. PLOS defines a study's minimal data set as the underlying data used to reach the conclusions drawn in the manuscript and any additional data required to replicate the reported study findings in their entirety. All PLOS journals require that the minimal data set be made fully available

Unfortunately, it is a legal restriction to provide the free access of the data. We have to keep the initial explanation, all the data belong to Planet Youth Iceland: due to internal and ethical rules and regulations, the whole data set is not to be made available for public sharing/ open access; however, the reasonable requests for data will be analysed and other interested researchers may receive data like: the values behind the means, standard deviations and other measures reported; the values used to build graphs, labels etc.

It is included from line 173 to line 185

Answers to Reviewer 1:

Dear reviewer,

Thank you very much for your comments and critics, it has helped us very much to understand the weakness of our article and to increase its scientific value.

1. It is not clear in the abstract the statistical analyzes that were carried out and what was presented as the objective of the study ("identify socio-cultural factors that may operate predictors of alienation and of deviant or even delinquent behavior"), is not aligned with what was presented as the main result ("Bucharest young people do not exhibit deviant behaviors or only to a minor degree, and that this is largely determined by a positive environment in terms of social relations, economic status, and level of culture.") . Furthermore, what is presented as the main result in the abstract is not congruent with the data presented in the manuscript, since 20% of the sample presented a type of behavior classified as “delinquency” and 12% as “high deviance” (Table 8).

We have rewritten the abstract in accordance with this observation.

2. In the introduction, the concepts of anomie and alienation are well defined and with a presentation of previous literature. Despite this, other variables that are addressed in the article (the risk factors assessed, such as parental monitoring, peer support, substance abuse, etc.) are not discussed based on the literature.

We have extended the introduction by including the other variables used in the analysis.

3. The method would require a characterization of the participants (age and gender) and a presentation of the psychometric parameters of the instrument adopted. In addition, more detail on the statistical analysis used would be necessary, describing the complete analysis process (data processing, choice of variables, descriptive statistics performed, procedure for carrying out logistic regression). As there is no description of the analysis procedure, it is not possible to know and to assess the rigor used in the process.

We have included the socio-demographic profile of the sample and we have remade the statistics, in a clearer way: two dependent variables, anomie and deviant behaviour, with several independent variables from various socio-cultural factors included in the survey. We hope it is more adequate now.

4. In the results, there is a presentation of definitions of scale variables and their composition that should be presented as “measures” in the method. There is no alignment between what is proposed to analyze the method and the analyzes presented in the results.

Indeed, we have moved these scales (and two other new ones: the level of anomie and time spent on leisure activities on the Internet) to the Method section. We have adjusted the research objective to the results. We have rewritten almost completely the Results section.

5. In the discussion, there is not enough discussion with literature data about the results found and the conclusions are not aligned with the results presented.

We have extended the discussion and we have included the relevant literature data.

ANSWERS TO REVIEW 2

First of all, thank you very much for your so friendly review and so emphatic approach. Please receive our deep gratitude for this!

1) I suggest that the title, “Transition from anomie to delinquency in adolescence: Profiles and associated factors”, be reconsidered, because the study does not exactly reflect the profile of young people, but rather the factors that associate the transition from anomie to delinquency.

Indeed, we have reformulated the title to be more appropriate to the study main idea. The new title is: Social predictors of the transition from anomie to deviance in adolescence. Case study: the 2018 Youth Planet research in Bucharest, Romania

2) In the Introduction, the authors refer to some studies that correlate the predictors of what could potentially trigger a transition from anomie to delinquency; however, none of them discuss the social or structural conditions that potentiate such situation. In developing countries, for instance, macroeconomic conditions prevail over individual and even community conditions regarding the experience of unruly and deviant adolescence. I believe it is important to emphasize this comparative aspect, which would strengthen the study and allow its scope to be expanded beyond subjective conditions and personal inabilities to comply or not with societal rules.

We have included in the Introduction some aspects related to juvenile delinquency in Romania.

3) This study also sought to respond to the following question: “What are the most relevant factors associated with juvenile delinquency in Bucharest, Romania?” To this end, the authors resourced to a 17-item questionnaire. To better contextualize the research, I suggest that the authors include some information, if available, on the state of the art of juvenile delinquency in Romania and, in particular, in Bucharest, because only the data presented do not show whether delinquency is high or low among young Romanians.

We have included in the Conclusions section some statistics related to the juvenile delinquency in Romania.

4) The Methodology does not clearly present the profile of the study participants: there is need to explain, for example, the profile of the young people who responded to the questionnaire: age group, sex and gender identity, social class, ethnicity, in order to better understand the eligibility criteria of the study.

We have included the socio-demographic profile of the sample.

5) The Results and their statistical analysis faithfully demonstrate and answer the research question.

We have almost completely remade the statistics and rewritten the Results section accordingly. 

6) Regarding the Discussion, there is need for a more comprehensive analysis of both the results found and the findings of the literature that support the arguments. How does the theory of anomie, proposed by Merton (1968), assists us with thinking about the transition from anomie to delinquency in young Romanians? What pre-existing conditions do these young people have that motivate them to go beyond culturally and socially determined limits? Making good use of the studies and theories presented here could better support the evidence of the results.

We have made some adjustment of the Discussion section in accordance with the results from the upgraded statistics; we know that there are still some aspects that can be improved and, if you consider it necessary, we can go deeper with the theoretical contextualisation.

7) In addition, the authors state that the amount of time spent on communication and information technologies (CIT) by today's youth emerges as a predictor of the high level of deviant behavior, but not necessarily of delinquent acts. To support this argument, it is important to review communication theories on the excessive and abusive use of CTI among young people.

We have detailed the negative consequences of the abusive use of NICT (especially of accessing the online universe from one’s personal mobile phone without adult monitoring).

---

## [Decision Letter · Decision Letter 1]

18 May 2022

Social predictors of the transition from anomie to deviance in adolescence

PONE-D-21-27300R1

Dear Dr. NADOLU,

We’re pleased to inform you that your manuscript has been judged scientifically suitable for publication and will be formally accepted for publication once it meets all outstanding technical requirements.

Kind regards,

Pedro Vieira da Silva Magalhaes, M.D., Ph.D.

Academic Editor

PLOS ONE

Additional Editor Comments (optional):

Reviewers' comments:

Reviewer's Responses to Questions

**Comments to the Author**

1. If the authors have adequately addressed your comments raised in a previous round of review and you feel that this manuscript is now acceptable for publication, you may indicate that here to bypass the “Comments to the Author” section, enter your conflict of interest statement in the “Confidential to Editor” section, and submit your "Accept" recommendation.

Reviewer #1: All comments have been addressed

Reviewer #2: All comments have been addressed

2. Is the manuscript technically sound, and do the data support the conclusions?

Reviewer #1: Yes

Reviewer #2: Yes

3. Has the statistical analysis been performed appropriately and rigorously? 

Reviewer #1: Yes

Reviewer #2: Yes

4. Have the authors made all data underlying the findings in their manuscript fully available?

Reviewer #1: Yes

Reviewer #2: Yes

5. Is the manuscript presented in an intelligible fashion and written in standard English?

Reviewer #1: Yes

Reviewer #2: Yes

6. Review Comments to the Author

Reviewer #1: First, I would like to emphasize and congratulate the extensive review work that the authors have carried out. I consider that all the requested changes have been made and that the work is ready for publication.

Reviewer #2: The manuscript presents, after peer review, adequacy to publishable standards. In addition, new elements were incorporated into the text, which facilitates the understanding of the study. I'm a favor of publication.

7. PLOS authors have the option to publish the peer review history of their article (what does this mean?). If published, this will include your full peer review and any attached files.

Reviewer #1: No

Reviewer #2: No

---

## [Editor Report · Acceptance letter]

13 Jun 2022

PONE-D-21-27300R1 

Social predictors of the transition from anomie to deviance in adolescence 

Dear Dr. NADOLU:

I'm pleased to inform you that your manuscript has been deemed suitable for publication in PLOS ONE. Congratulations! Your manuscript is now with our production department. 

Kind regards, 

on behalf of

Professor Pedro Vieira da Silva Magalhaes 

Academic Editor

PLOS ONE